# Seasonal Oxy-Inflammation and Hydration Status in Non-Elite Freeskiing Racer: A Pilot Study by Non-Invasive Analytic Method

**DOI:** 10.3390/ijerph20043157

**Published:** 2023-02-10

**Authors:** Andrea Brizzolari, Gerardo Bosco, Alessandra Vezzoli, Cinzia Dellanoce, Alessandra Barassi, Matteo Paganini, Danilo Cialoni, Simona Mrakic-Sposta

**Affiliations:** 1Environmental Physiology and Medicine Laboratory, Department of Biomedical Sciences, University of Padova, 35131 Padova, Italy; 2DAN Europe Research Division, 64026 Roseto degli Abruzzi, Italy; 3Institute of Clinical Physiology, National Research Council (IFC-CNR), Piazza dell’Ospedale Maggiore, 3, 20162 Milan, Italy; 4Department of Health Sciences, Università degli Studi of Milan, 20142 Milan, Italy

**Keywords:** oxidative stress, inflammation, saliva, urine, mountain, skiing, training, electron paramagnetic resonance

## Abstract

Freeskiing is performed in an extreme environment, with significant physical effort that can induce reactive oxygen species (ROS) generation and dehydration. This study aimed to investigate the evolution of the oxy-inflammation and hydration status during a freeskiing training season with non-invasive methods. Eight trained freeskiers were investigated during a season training: T0 (beginning), T1-T3 (training sessions), and T4 (after the end). Urine and saliva were collected at T0, before (A) and after (B) T1-T3, and at T4. ROS, total antioxidant capacity (TAC), interleukin-6 (IL-6), nitric oxide (NO) derivatives, neopterin, and electrolyte balance changes were investigated. We found significant increases in ROS generation (T1A-B +71%; T2A-B +65%; T3A-B +49%; *p* < 0.05–0.01) and IL-6 (T2A-B +112%; T3A-B +133%; *p* < 0.01). We did not observe significant variation of TAC and NOx after training sessions. Furthermore, ROS and IL-6 showed statistically significant differences between T0 and T4 (ROS +48%, IL-6 +86%; *p* < 0.05). Freeskiing induced an increase in ROS production, which can be contained by antioxidant defense activation, and in IL-6, as a consequence of physical activity and skeletal muscular contraction. We did not find deep changes in electrolytes balance, likely because all freeskiers were well-trained and very experienced.

## 1. Introduction

Freeskiing is a specific type of alpine skiing, where athletes twist and throw the bodyweight in addition to the weight of the equipment, leading to a rapid transition of the leg tension that propels the athletes and allows for tricks and rotations in the vertical and horizontal planes of rotations [1]. Physical exercise is a well-known source of reactive oxygen species (ROS) in response to raised O_2_ utilization to sustain the effort, especially in skeletal muscles and, to a lesser extent, in several tissues, including heart, lungs, or blood [2]. ROS play a key role in the modulation of cell redox status and signaling processes, and they are involved in the regulation of several patho-physiological mechanisms [3,4]. Nevertheless, a prolonged duration as well as a high intensity of endurance training can lead to increased ROS production, possible macromolecule damage, and, thereby, to oxidative stress [5,6,7,8], compromising the efficacy of immune system and inflammatory response [9]. Furthermore, oxidative stress might induce programmed cell death, typically through intrinsic apoptotic pathways [10]. These effects can damage the skeletal muscle, reducing physical performance [11].

In alpine ski athletes, the imbalance between pro- and antioxidant species is a consequence of the cumulative effect of intensive exercise and hypoxia at moderate altitude [12,13]. Some authors found that oxidative stress peaks during the midseason [14], with an increase in uric acid and decrease in oxidized low-density lipoproteins (LDL) [15]. In addition to ROS production, an increase in nitric oxide (NO), a signaling molecule involved in response to acute sessions of exercise to adapt the body to exercise training [16], can be found under stressful conditions [17]. NO can react with superoxide anion, giving rise to aggressive reactive nitrogen species (RNS) [18], exacerbating the oxidative stress.

It is well known that physical training improves the efficiency of the endogenous antioxidant system, reducing exercise-induced oxidative stress [2,3,19,20]. However, prolonged training sessions can reduce the efficiency of antioxidant defenses, leading to an excessive production [4,5,21,22].

Environmental conditions, such as cold and dry air, can lead to raised respiration water loss, reduced thirst perception, and increased diuresis [23,24]. This cold-related diuresis seems to be the consequence of raised atrial naturetic peptide secretion and/or inhibition of antidiuretic hormone release [25], accelerating free-water clearance and increasing urine volume, which, if left unchecked, impairs fluid balance [26]. If the fluid intake cannot balance, dehydration occurs, causing a state of hypohydration. During races, athletes often avoid drinking much while are on the slopes for the practical reason of not worsening their hydration status [23]. Since both hypohydration and dehydration may compromise performance, re-hydrating during recovery is equally important because a decreased intracellular fluid volume is reported to impair glycogen and protein resynthesis [27,28]. 

The aims of this study were to examine, for the first time, the freeskiers athletes and, moreover, with non-invasive methods [29,30], to examine the evolution of the oxy-inflammation and hydration status during a training season of these athletes. Freeskiing, is a sport performed in an extreme environment, with significant physical effort, and combined with other factors, can induce ROS production and electrolyte imbalance. 

The method herein presented allows for reliable, simple, and non-invasive measurements (saliva and urine samples) in order to screen and monitor the subjects in the field. This method can become a routine test carried out for sample collection by not only professional but also non-professional athletes. 

## 2. Materials and Methods

### 2.1. Subjects

The Prato Nevoso freeskiers team, eight trained male experts (mean: age 25.1 ± 6.1; height 179.5 ± 7.0 cm; weight 72.4 ± 7.3 kg; and BMI 22.5 ± 2.4) were investigated during a training season. Volunteers were not professional athletes and trained from Friday to Sunday, as they were busy with their respective jobs during the week. During the week, they sporadically practiced other sports, including cycling, indoor climbing, surfing, and skating. 

During the first session (T0), a questionnaire was administered to collect data on the subjects’ training experience. On average, the subjects had 11.3 ± 9.8 years of alpine skiing and freeskiing experience. 

Blood pressure was recorded in triplicate by an automatically inflating cuff around the upper arm (Omron M7, Intelli IT, Omron, Japan), and peripheral arterial oxygen saturation (SpO_2_) was measured using a pulse oximeter (TuffSat, GE Datex Ohmeda, Colcester, CT, USA).

No subject reported previous episodes of Mountain Acute Sickness, historical or clinical evidence of arterial hypertension, cardiac, pulmonary, or any other significant disease; none of them took prescription drugs, suffered any acute disease during the 15 days before the experiment, reported consumption of anti-inflammatory drugs in the 7 days before the experiment, or performed intense physical effort in the 7 days before any training session. All athletes were non-smokers and in good health.

All the athletes received an explanation of the study’s purposes, risks, and benefits, were familiarized with the experimental protocol, and read and signed a specific informed consent form before the experiment. The study was conducted in accordance with the Helsinki Declaration and was approved by the Ethical Committee of Università degli Studi di Milano, Italy (Aut. n 37/17).

### 2.2. Experimental Design

This observational study was carried on during the ski season, from 8 December 2021, to 17 April 2022, at the Prato Nevoso Ski resort, at 1480 m (Frabosa Sottana, CN, Italy), see Figure 1. Freeskiers performed their training sessions in the Prato Nevoso snow park, an area of the ski resort track or ski trail, created for snowboarders and freeskiers to perform tricks.

Biological samples were collected, and measurements were carried out as follows:In December (T0), before the season beginning. These were basal measures;In February (T1), at the start of training session;In March (T2), during the middle period of the season;In April (T3), at the end of the season;In June (T4), two months after the end of the season.

All the sessions were performed on Saturdays.

#### Training Exercise Session

As shown in Figure 1, from T1 to T3, the freeskiing athletes were tested pre- and post-training session. Daily training consisted of 4 h (9.00 a.m.–1.00 p.m.), distributed as follows:9.00–9.15 a.m.: dry land warm-up (stretching) before going to ski run;9.15–10.00 a.m.: warm-up on skin run (flat tricks);10.00–10.15 a.m.: speed and snow condition check;10.15–11.30 a.m.: riding in snow park, Part 1 (jumps, rails, jibbing, and slang);11.30–11.45 a.m.: break;11.45 a.m.–1.00 p.m.: riding in snow park, Part 2 (jumps, rails, jibbing, and slang).

During warm-up, subjects ran across the ski track close to the snow park three times, performing some flat tricks:Hand Drag 180, a 180° turn involving landing on the hands while the legs are up in the air; and Hand Drag 360, a full turn with the legs up in the air while landing on the hands.Nollie, a jump on a flat slope where skiers push from their noses; and Nollie 180, a jump similar to Nollie with a 180° turn.Nose Butter 360, a 360° rotation around the vertical axis with pressure being put on the noses; and Nose Butter 540, a 540° rotation around the vertical axis with pressure being put on the noses and back end.Switch Tail Butter 360, a 360° rotation around the vertical axis with pressure being put on the tails, performed from switch.

After warm-up, the rides in snow park were performed following a check of the snow consistency. In every training session, all the athletes respected the training plan and the timing. Freeskiers rode the snow park line of the Prato Nevoso Ski resort six times, with a break between the third and the fourth ride. 

The snow park line consists of an area of a ski trail with dedicated structures (jumps, rails, jibbing, and slang), created for freeskiers to perform tricks; the snow park line includes 2 jumps, 2 rails, 2 slang, and 1 jibbing area, for a total length of 500 m:-1st 100 m: 2 Nose Butter 360 and 1 Nose Butter 540;-2nd 100 m: 3 Switch Tail Butter 360;-3th 100 m: 2 Nose Butter 360 and 2 Nose Butter 540;-4th 100 m: 2 Nollie and 2 Nollie 180;-5th 100 m: 2 Switch Tail Butter 360, 1 Hand Drag 180, and 1 Hand Drag 360.

Chairlift takes 9 min to reach the top of the snow park. During the recorded training sessions, the air temperature ranged from −9 to −2 °C, wind from 11 to 13 km/h, and humidity from 57 to 85%.

### 2.3. Sample Collection

Approximately 1 mL of saliva was collected before (A) and after (B) of the training session using a Salivette device (Sarstedt, Nümbrecht, Germany). Before storage, saliva samples were centrifuged at 3000 rpm for 20 min. [31]. In addition, urine samples were collected by voluntary voiding in a sterile container before and after the training session. The sample collection at the pre-session (baseline) was always conducted at approximately 8.30 am.

All biological samples were initially stored at 4 °C in a portable cooler during the transport back to the laboratory, aliquoted, then stored at −20 °C until assayed, and thawed only once before analysis.

### 2.4. Biomarkers Assessment

#### 2.4.1. Reactive Oxygen Species (ROS) and Total Antioxidant Capacity (TAC)

An X-band electron paramagnetic resonance spectroscopy (EPR, 9.3 GHz) (E-Scan, Bruker Co., Billerica, MA, USA) was used to detect ROS production and TAC in saliva samples. Methods were previously described [29,30,32,33,34]; briefly, spin probe CMH (1-hydroxy-3-methoxy-carbonyl-2,2,5,5-tetramethylpyrrolidine) was used for ROS determination, and a stable radical CP (3-carboxy2,2,5,5-tetramethyl-1-pyrrolidi-nyloxy) was used as an external reference to convert ROS determinations into absolute quantitative values (μmol·min^−1^). Spin–trap 1,1-diphenyl-2-picrylhydrazyl (DPPH•) was used to measured TAC. The calculated antioxidant capacity was expressed in terms of Trolox equivalent antioxidant capacity (TAC, mM). 

All samples were stabilized at 37 °C using a Temperature Controller unit (Noxigen Science Transfer & Diagnostics GmbH, Elzach, Germany), interfaced with the spectrometer.

#### 2.4.2. Interleukin-6 (IL-6)

IL-6 urinary levels were determined by ELISA kit (Cayman Chemical, Ann Arbor, MI, USA, Item No. 501030), according to the manufacturer’s instructions. The determinations were assessed in duplicate, and the inter-assay coefficient of variation was in the range indicated by the manufacturer.

#### 2.4.3. NO Metabolites

NO derivatives, nitrate and nitrite (NO_2_ + NO_3_ = NOx), were measured in urine samples by a colorimetric method based on the Griess reaction, using a commercial kit (Cayman Chemical, Ann Arbor, MI, USA) previously described [35]. Samples were read at 545 nm, and the concentration was assessed by a standard curve. 

#### 2.4.4. Creatinine, Uric Acid, and Electrolytes

Creatinine, urea, uric acid, sodium (Na^+^), potassium (K^+^), chlorine (Cl^−^), magnesium (Mg^2+^), phosphorus (P), Calcium (Ca^2+^), and total proteins were investigated in urine samples. Urine samples were defrizzed, at room temperature, shacked with the Vortex for 5 s for homogenization before the analysis. Briefly, 500μL of each sample was placed in a plastic test tube. The test tubes were allocated in the auto sampler of a Roche Cobas^®®^ 6000 analyzer (Roche Diagnostics, Basel, Switzerland). The reported total imprecision was <2.8%, while the intra assay CV% was <1.8%.

#### 2.4.5. Neopterin

Neopterin urinary concentrations were measured by high-performance liquid chromatography (HPLC) method, as previously described [33,36]. The calibration curve was linear over the range of 0.125–1 µmol/L. Inter-assay and intra-assay coefficients of variation were <5%.

#### 2.4.6. Urine Test Strip

The Urine Test Strip (Siemens Healthcare S.r.l. 10sys Multistix, Italy) was used for semi-quantitative determinations of pH, urobilinogen, bilirubin, ketones, specific gravity/density, and leukocytes in urine. Test was immediately performed after sample collection in duplicate for each subject.

### 2.5. Scale for Assessment of Physical Fatigue and Recovery

The subject-perceived exertion was assessed immediately after every training session on the basis of the physical sensations, and muscle fatigue was assessed by the Borg Rate of Perceived Exertion scale (RPE) [37].

The quality of training recovery was assessed at pre-session training and compared with the last training by Total Quality of Recovery scale (TQR) proposed by Kenttä and Hassmén [38].

To study the subjective mood, general wellness (happy/unhappy, rested/tired), general sensation (hot/cold, calm/agitation), and head no pain/pain were evaluated using a 0–100 mm visual analog scale (VAS) to test the subjective perception [39]. If pain was present, its location was requested. 

The Profile of Mood States (POMS) is a popular tool among sport psychologists who have used it to compare the prevailing moods of elite athletes and non-athletes. In short-form, this tool was administered to measure certain psychological traits, focusing on tension–anxiety, depression–dejection, anger–hostility, vigor–activity, fatigue–inertia, and confusion–bewilderment on a five-point scale from 0 to 4 [40,41,42]. High vigor scores reflect a good mood or emotion, and low scores in the other subscales reflect a good mood or emotion.

### 2.6. Statistical Analysis

Analysis was performed using the GraphPad Prism package (GraphPad Prism 9.5.0, GraphPad Software Inc., San Diego, CA, USA) and SPSS statistics software (IBM corporation). Data are presented as mean ± SD. Statistical analyses were performed using non-parametric tests; Wilcoxon matched-pairs signed-rank test for independent samples due to the small sample size for comparing pre- vs. post-training session; and ANOVA repeated measures, with Dunn’s multiple comparison tests to further check the among-groups significance. A *p* < 0.05 was considered statistically significant. dCohen with 95% CI was used for calculating the size effect. Moreover, we used the Hopkins scale for classification of the effect size. Change Δ% estimation (((pre value − post value)/pre value) × 100) is also reported in the text.

## 3. Results

Physiological parameters obtained at the different time-points during the season are reported in Table 1. At the end of the season, a significant (*p* < 0.05) decrease in systolic blood pressure was recorded, dCohen = 0.97.

The concentration values of oxy-inflammation biomarkers obtained during the training sessions (A pre vs. B post; T1, T2, and T3) and from T0 and T4 in the examined athletes are displayed in Figure 2.

We found a significant increase in ROS production after every training session: T1A vs. B +71%, dCohen = 1.10; T2A vs. B +65%, dCohen = 1.28; T3A vs. B +49%, dCohen = 1.23; mean +61%, *p* < 0.05–0.01 (Figure 2A), and in IL-6 values after training sessions: T2A vs. B +112%, dCohen = 1.66; T3A vs. B +133%, dCohen = 2.01; mean +128%, *p* < 0.01; (Figure 2C), while a decrease in TAC was found at training sessions: T1A vs. T1B: −17%, dCohen = 1.53; (*p* < 0.05 (Figure 2B). NO metabolites did not show any statistical difference (Figure 2D).

Furthermore, ROS and IL-6 showed statistically significant differences between T0 and T4 (*p* < 0.05): ROS +48%, dCohen = 1.09; IL-6 +86%; dCohen = 1.22; while TAC only between T0 and T3A: −12%, dCohen = 1.23; (*p* < 0.05). 

The values of urea, uric acid, Na, K, Cl, P, Mg, Ca, and total proteins observed during training sessions T0 through T4 are reported in Table 2. Urea and uric acid exhibited statistically significant differences between T0 and T4 dCohen = 0.60 and dCohen = 1.61, respectively. K showed statistically significant differences in post session value T2B (+53%) vs. T2A dCohen = 0.91, while P at T3B vs. T3A (+94%) dCohen = 0.87.

Moreover, as reported in Figure 3, we found significant changes in neopterin levels after every training session: T1A vs. B +86%, dCohen = 1.76; T2A vs. B + 44%, dCohen = 1.29; and T3A vs. B +46%, dCohen = 1.26; *p* < 0.05.

Urine standard parameters are reported in Table 3. No significant differences were found during the season.

In Figure 4 are reported significant changes in Borg and TQR scores. No significant differences are reported intra-season for measuring physical activity intensity level (Figure 4A), with a mean of 12–13 score during the season, which corresponded to HR max 65–70% and a VO_2_ max 50–75%; on the other hand, significant differences (*p* < 0.05–0.01) were found intra-season in the TQR scores at T1, T2, and T3 with respect to T0, dCohen = 1.26.

Finally, no significant differences were found in total mood score of POMS scale and in VAS item score during a single training session (A vs. B) or during the entire training season. Particularly, we observed percentage changes in the general wellness measure with respect to T0: T1A +79%, T2A +100%, T3A −47%, and T4A −2% (Figure 4C); the pain measure with respect to T0: T1A −10%, T2A −25%, T3A −65%, and T4A −65% (Figure 4D); and the anxiety measure with respect to T0: T1A −12%, T2A −40%, T3A −96%, and T4A −33%.

In order to understand the results obtained and then evaluate the effects of training vs. seasonality in freeskiers, we applied the Hopkin scale, as shown in Figure 5. Significant group differences and large effect sizes were found for uric acid, TAC, and IL-6 during the season, while moderate effect sizes were found for ROS, SBP, and urea at the end of the freeski season. Significant group difference and a very large effect size was found for IL-6 after the T3 training session, and large effect sizes were found for neopterin, IL-6, TAC, ROS, and TQR during the training session. Finally, moderate effect sizes were found for ROS at T1, K at T2, and P at T3 training sessions.

## 4. Discussion

This was the first study that attempted to investigate the evolution of oxy-inflammation and hydration parameters among freeskiers during a training season, and importantly, to do so by noninvasive measurements.

Exercise-induced ROS generation is typical of prolonged and/or high training loads, such as those of endurance activities [36,43,44,45]. In muscle fiber during physical effort, the rate of ROS generation, such as O_2_•, is increased, influencing the fiber contraction [46]. An excess of ROS in skeletal muscle results in a decrease in the ability to generate force, leading to a fatigue condition [47].

We observed a significant increase in ROS production after every training session as a result of the physical effort. Importantly, ROS production plays a key role in cell signaling pathways involved in muscle adaptation to effort [48] with multi-protein pathways and signaling [49]. Furthermore, we found an increase of ROS basal value at T3A and at the end of the training season (T4), likely due to a decrease in the efficiency of the endogenous antioxidant defenses and a rising soreness. Athletes can activate the endogenous antioxidant system to control exercise-induced oxidative stress, avoiding the macromolecules damage [50,51,52,53,54]. A significant decrease in TAC was measured only after the first training session (T3B), while other sessions did not exhibit significant change, despite a reduction of TAC level. During the first training session, the TAC value was higher than during the other sessions, resulting in a higher starting antioxidant defense level. This may explain the significant training-induced decrease in TAC value, attributable to a higher baseline, as observed by some authors [55]. Training may also induce a converging of TAC values towards an optimal level, adapting the body to the exercise-related fatigue and leading to a smaller variation after the successive session. 

IL-6 is produced mainly within the working skeletal muscles [56], with its released amount strongly related to the exercise duration and intensity as well as the mass of muscle recruited and endurance capacity [57,58]. We found raised IL-6 values after physical exercise in every training session, in accordance with other authors [59]. IL-6 increases significantly after eccentric exercise [60] in response to long-duration exercise, independently of muscle damage. IL-6 release in relation to muscle fiber damage occurs later and is of a smaller magnitude than IL-6 production related to muscle contractions [57]. Skiing requires a lot of eccentric contractions that occur when the muscle lengthens at the same time it contracts. In our case, freeskiing implies muscle eccentric contractions associated with concentric contraction during jump execution. 

We did not find any difference in NOx level after any of the training sessions. In our case, we did not introduce any NO_2_ or NO_3_ supplementation to the athletes. NO derivatives may improve exercise tolerance, reducing O_2_ consumption and optimizing muscle contraction efficiency [61,62] through their role in the regulation of blood flow, contractility, glucose and Ca^2+^ homeostasis, mitochondrial respiration, and biogenesis [63]. 

Despite the increase, uric acid did not change significantly after any of the sessions, similar to the data of some authors [64]. Exercise-related changes of uric acid are contrasting. During intense exercise, uric acid excretion is reduced [65,66] as a consequence of an antidiuretic hormone increase [67] and an elevated lactate production, which leads to a decrease in the urinary excretion of oxypurines [68,69]. On the other hand, other research groups found an increase of urinary uric acid [70,71], which may be the consequence of purine release from the muscle after prolonged exercise rather than a decrease in plasma purine removal rates [72]. In muscle, ATP consumption can trigger a cascade of nucleotide degradation to purine catabolic intermediates that increase uric acid level after physical activity [65].

We evaluated the hydration status of the freeskiers without any drink supplementation. We did not find deep changes in electrolytes balance, likely because all volunteers were well-trained and very experienced. Urea value did not change significantly after any of the training sessions, likely to reduce exercise-related water loss. Urea is reabsorbed during passage of the filtrate through the tubule of the nephron, activating a mechanism of water-conserving action [73].

Despite their elevated values, Na and K levels did not show significant variation during the season, with the exception of K variation after the second training session (T2B). K is involved in the regulation of osmotic pressure and plays an important role in nerve stimulation and muscle contraction during exercise [74]. 

Urinary P decreased after every training session, significantly after the third session (T 3B). This phosphorus excretion reduction may reflect a slight respiratory alkalosis caused by hyperventilation during exercise [75]. 

Cl, Mg, and Ca did not change significantly during the season. Our findings seem to indicate freeskiing might be not intense enough to induce significant differences in electrolyte concentrations between urine samples taken before and after the physical activity. Furthermore, the absence of fluid intake may also be a possible reason for a major stability in concentration of circulating electrolytes, reducing their loss through urine [76]. 

Neopterin can rise during systemic oxidative stress, as reported by some authors [7,32,77]. In our freeskiers, neopterin increase can be associated with ROS production, similar to other studies [29,78].

Prolonged physical activity can impair kidney function as consequence of several physiological mechanisms to manage stress [7,33]. We did not find any difference in urine standards parameters, likely due to the good health and high level of experience of the subjects to adapt to the freeskiing-related physical effort.

We observed a progressive decrease in TQR score throughout the training season. This decrease may be due to the body adapting to exercise-induced fatigue, which has a beneficial effect on the athletes’ psychological condition [79]. Freeskiers showed an increase in general wellness sore and a decrease in pain score at the end of the training season, evidence of the fact that physical activity improves the psychophysical status of athletes. 

Indeed, it is well known that physical activity can improve mental health, reducing symptoms of psychological distress, such as depression, anxiety [80,81], failure after competition [82], and increasing self-esteem and cognitive functions [83].

## 5. Limitation

The main limitation of our study is the small sample size. We included the entire team of Prato Nevoso freeskiers. It should be emphasized that the freeski is not a very widespread sport, unlike alpine skiing. Therefore, results must be interpreted carefully. In fact, we studied both young and non-professional subjects. As this is a pilot study, we plan to pursue professional and non/professional and national/international athlete’s collaborations in the future to increase the number of subjects studied. Nevertheless, this is the first study to assess ROS production, antioxidant capacity, nitric oxide metabolites levels, inflammatory status, and electrolytes levels, in a freeskiers athletes in a real field environment.

## 6. Conclusions

Freeskiing induced an increase in ROS production that can be contained by antioxidant defense activation. Neopterin increased as a consequence of ROS generation, while IL-6 increased as a consequence of physical activity and muscular contraction type.

We did not find deep changes in electrolytes balance, likely because all subjects examined were well-trained and very experienced. Moreover, freeskiing-related physical activity can improve body adaptation to exercise-induced fatigue and mental health, reducing symptoms of psychological distress, including anxiety. Finally, despite a small sample size, the current findings show a large effect size so that this pilot study presumably may have a practical significance.

## Figures and Tables

**Figure 1 ijerph-20-03157-f001:**
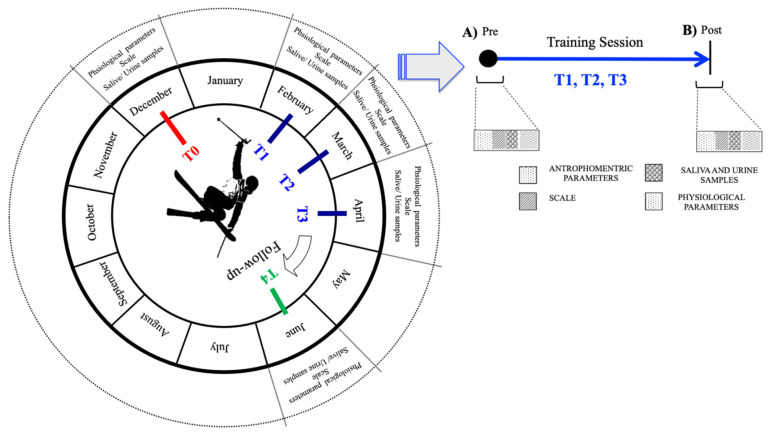
Study protocol design, seasonal circle, and session training. Freeskiing athletes were tested five times throughout a competitive season: at the beginning (T0); every month (T1, T2, and T3); and two months after the end of the season (T4). Furthermore, as shown in the scheme on the right of the figure, from T1 to T3, the athletes were tested pre- (**A**) and post- (**B**) training session.

**Figure 2 ijerph-20-03157-f002:**
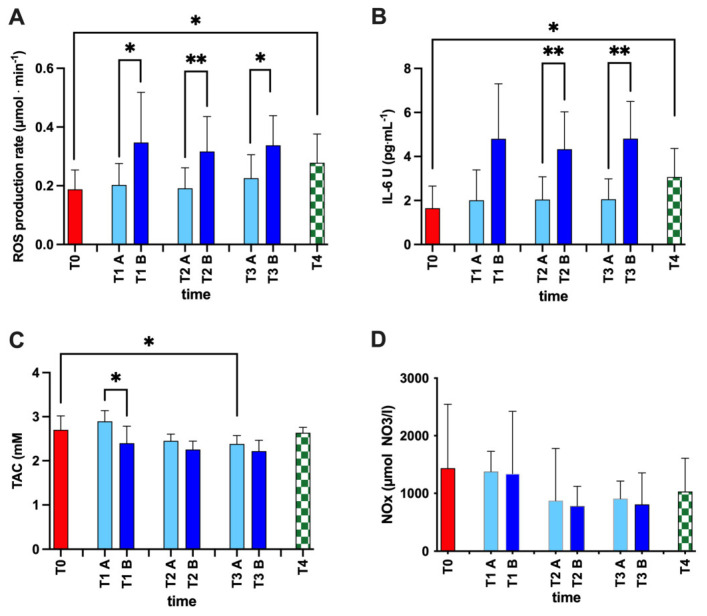
Histogram panel plots of the oxy-inflammation biomarkers. Time course of (**A**) reactive oxygen species (ROS), (**B**) interleukin (IL-6), (**C**) total antioxidant capacity (TAC), and (**D**) NO metabolites (NOx) at: T0, basal measure, before the season; T1, at the start of training season; T2, the middle period of the season; T3, the end of the season; and T4, two months after the end of the season. Values recorded from pre- (TA) and post- (TB) training sessions are also reported. Data are expressed as mean ± SD. Statistically significant differences comparison are displayed as: *, *p* < 0.05; **, *p* < 0.01.

**Figure 3 ijerph-20-03157-f003:**
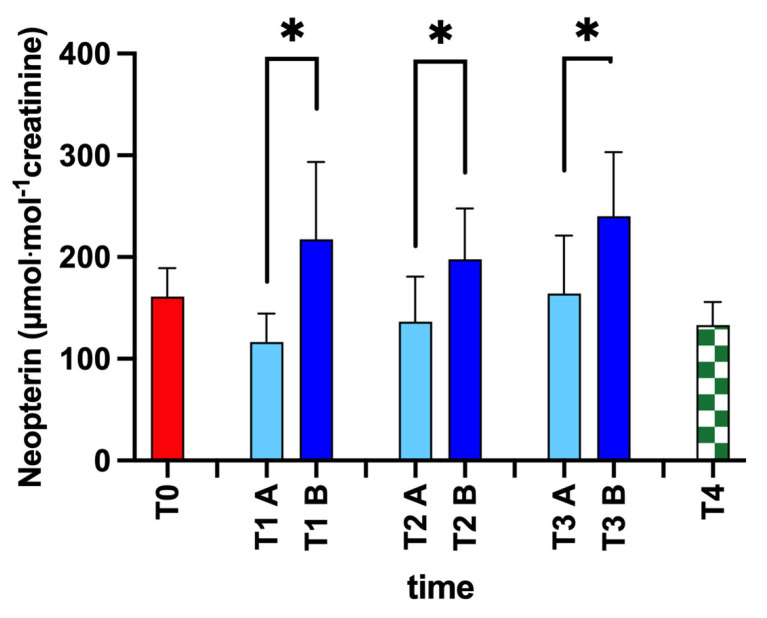
Neopterin changes during the training season. Statistically significant differences symbols: * *p* < 0.05.

**Figure 4 ijerph-20-03157-f004:**
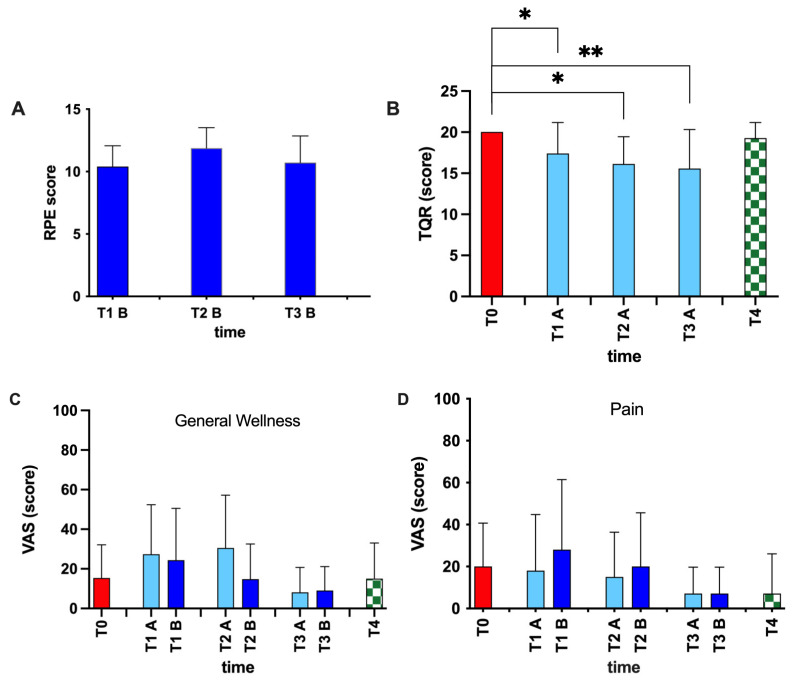
Borg and TQR scales and VAS score from all subjects. Borg scale (**A**) was assessed after every training session, while TQR scale (**B**) was assessed at the baseline (T0), before every training session, and after the season (T4). VAS scores for general wellness (**C**) and pain (**D**) were assessed at the baseline (T0), before (**A**) and after (**B**) every training session, and after the season (T4). Statistically significant difference comparisons are displayed as: * *p* < 0.05; ** *p* < 0.01.

**Figure 5 ijerph-20-03157-f005:**
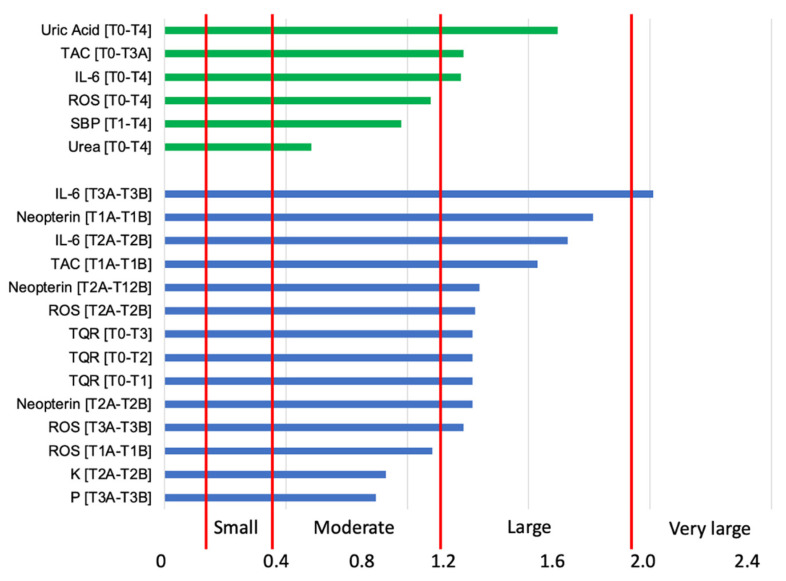
Relative difference among selected measures in athletes during season (green bars) and during training session (blue bars).

**Table 1 ijerph-20-03157-t001:** Physiological parameters from all subjects at the baseline (T0) and during the season: SBP: Systolic Blood Pressure; DBP: Diastolic Blood Pressure; and SpO_2_: Arterial Oxygen Saturation. Data are displayed as mean ± SD. Statistically significant differences symbols: * *p* < 0.05 respect T0. The colors of the time boxes (from T0 to T4), are the same as those reported in the histograms of figures from 2 to 4, to help the reader.

Parameter	Time Point				
	T0	T1	T2	T3	T4
**SpO_2_** **(%)**	97.0 ± 0.9	97.4± 1.1	97.3± 1.1	97.1± 1.0	97.5± 1.3
**SBP** **(mmHg)**	135.6 ± 24.9	120.0 ± 7.3	116.9 ± 16.4	117.9 ± 5.8	116.5 ± 12.1 *
**DBP** **(mmHg)**	79.5 ± 11.3	76.8 ± 5.4	65.2± 5.9	67.6± 8.7	74.0 ± 7.5

**Table 2 ijerph-20-03157-t002:** Urea, uric acid, electrolytes (Na, K, Cl, P, Mg, and Ca), and total proteins at the baseline (T0), during (T1-T3), and after the season (T4). Data are displayed as mean ± SD. Statistically significant differences symbols: * *p* < 0.05 respect T0; ^†^
*p* < 0.05, ^††^
*p* < 0.01 post (B) respect to pre (A). The colors of the time boxes (from T0 to T4), are the same as those reported in the histograms of figures from 2 to 4, to help the reader.

	Time Point							
	T0	T1		T2		T3		T4
		A	B	A	B	A	B	
**Uric Acid (mg/dL)**	9.3 ± 4.6	16.4 ± 18.5	32.1 ± 26.1	21.8 ± 25.0	30.4 ± 19.5	25.6 ± 15.1	23.4 ± 14.3	41.9 ± 28.1 *
**Urea (mg/dL)**	984.0 ± 346.3	1555.0 ± 362.7	1225.0 ± 184.0	1058.0 ± 255.4	1040 ± 373.4	1426.0 ± 386.3	1028.0 ± 546.1	736.9 ± 351.6 *
**Na (mmol/L)**	106.6 ± 39.7	144.8 ± 17.7	191.6 ± 59.7	153.6 ± 60.5	198.6 ± 50.4	148.3 ± 64.1	139.9 ± 95.7	131.7 ± 68.2
**K (mmol/L)**	61.5 ± 30.6	62.9 ± 22.7	95.7 ± 36.9	60.5 ± 35.1	92.0 ± 34.0 ^††^	40.0 ± 9.0	43.7 ± 20.6	62.7 ± 38.1
**Cl (mmol/L)**	124.8 ± 43.5	122.0 ± 44.1	204.0 ± 73.9	157.0 ± 78.7	224.1 ± 50.1	125.4 ± 53.1	138.0 ± 81.6	149.0 ± 58.3
**P (mg/dL)**	106.3 ± 58.6	142.3 ± 71.3	66.6 ± 41.2	81.1± 36.6	50.8 ± 20.4	105.7 ± 51.6	54.2 ± 65.6 ^†^	53.2 ± 35.1
**Mg (mg/dL)**	5.0 ± 5.1	15.2 ± 8.7	6.7 ± 4.2	8.4 ± 3.2	6.3 ± 3.1	11.9 ± 5.1	7.9 ± 7.4	5.8 ± 3.0
**Ca (mg/dL)**	7.9 ± 8.0	16.0 ± 5.8	10.2 ± 7.9	13.3 ± 5.0	11.6 ± 6.0	15.4 ± 6.6	10.7 ± 7.1	9.8 ± 5.1
**Total proteins (mg/dL)**	51.3 ± 3.5	50.0 ± 1.0	81.2 ± 69.7	50.0 ± 1.0	50.0 ± 1.0	50.0 ± 1.0	70.6 ± 30.8	66.7 ± 19.4

**Table 3 ijerph-20-03157-t003:** Urine standard analysis. Urine test strip data values (mean ± SD) from all subjects. The colors of the time boxes (from T0 to T4), are the same as those reported in the histograms of figures from 2 to 4, to help the reader.

	Time Point				
	T0	T1	T2	T3	T4
**Bilirubin (μmol·L^−1^)**	4.3 ± 12.37	3.1 ± 7.8	3.9 ± 7.5	4.1 ± 6.6	4.9 ± 8.7
**Urobilinogen (μmol·L^−1^)**	0.1 ± 0.07	0.2 ± 0.2	0.2 ± 0.0	0.2 ± 0.0	0.1 ± 0.8
**Ketones (mmol·L^−1^)**	1.0 ± 5.3	1.3 ± 4.4	2.5 ± 2.5	2.9 ± 3.6	2.6 ± 3.3
**pH**	5.9 ± 0.6	6.3 ± 0.5	6.3 ± 0.5	6.2 ± 0.5	6.2 ± 0.5
**Leucocytes (Leuko·μL^−1^)**	0.0 ± 0.0	0.0 ± 0.0	0.0 ± 0.0	0.0 ± 0.0	0.0 ± 0.0
**Specific gravity**	1.02 ± 0.0	1.02 ± 0.0	1.02 ± 0.0	1.02 ± 0.0	1.02 ± 0.0

## Data Availability

Data are available at request from the authors.

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
