# Peer review of "Seasonal Oxy-Inflammation and Hydration Status in Non-Elite Freeskiing Racer: A Pilot Study by Non-Invasive Analytic Method"

_ijerph, 2023, doi:10.3390/ijerph20043157_

Round 1

Reviewer 1 Report

GENERAL COMMENTS.

In general, the manuscript purpose is interesting, i.e., (i) to examine the evolution of the oxy-inflammation and hydration status during a training season with non-invasive methods. However, I have some questions regarding statistical analysis. First, the size of sample is very restrict to confirm this results, and secondly, I suggest to the authors to change the resolution of figures of at least 300 dpi

 SPECIFIC COMMENTS

Introduction. Specify the rational of the study and the hypothesis has to be clarified.

 Methods

Participants: The major weakness of this study was the sample size. So please add the results of effect size and power simulation of this study (using G*power for example) to prove that this sample was sufficient to have 80% of statistical power. 

 Statistical analyses

-       Please add ICC

-       Please add effect sizes (ES) from ANOVA output by converting partial eta-squared to Cohen’s d.

 Results.

-       I suggest to the authors to add 1) the effect size for all interactions after converting partial eta-squared values to Cohen’s d values.

-       So it should be better to include the scale of Hopkins to interpret the results.

-       Poor resolution of figures. Please change the resolution of at least 300 dpi

 Discussion.

-       The discussion can be tightened up. As is, it lacks focus in many areas for example the specific purpose and findings of this study

-       More notes about practical application associated to the results of the study could be implemented. 

 Conclusion.

The conclusion to be reformulated (please try to make it shorter).

Author Response

GENERAL COMMENTS.
In general, the manuscript purpose is interesting, i.e., (i) to examine the evolution of the oxy-inflammation and hydration status during a training season with non-invasive methods. However, I have some questions regarding statistical analysis. First, the size of sample is very restrict to confirm this results, and secondly, I suggest to the authors to change the resolution of figures of at least 300 dpi.
Thanks for the review and suggestions.
We changed the figures resolution, all TIFF at 300dpi

SPECIFIC COMMENTS
Introduction. Specify the rational of the study and the hypothesis has to be clarified.
The aim of the study is now clarified.
Methods
Participants: The major weakness of this study was the sample size. So please add the results of effect size and power simulation of this study (using G*power for example) to prove that this sample was sufficient to have 80% of statistical power. 
It was not possible to calculate the sample size, because we studied all the athletes of the Prato nevoso freeskers team. This sentence has been added in the text in subjects 2.1 paragraph.

Statistical analyses

-       Please add ICC 

-       Please add effect sizes (ES) from ANOVA output by converting partial eta-squared to Cohen’s d.

Thank you. We added the part in statistical analysis (paragraph 2.6) and calculated the Cohen’s values.
Results.

-       I suggest to the authors to add 1) the effect size for all interactions after converting partial eta-squared values to Cohen’s d values. 

Thank you. We added the d-Cohen values in the results paragraph.
Taking into account that Cohen suggested that d = 0.2 be considered a 'small' effect size, 0.5 represents a 'medium' effect size and 0.8 a 'large' effect size. Our d-Coen values show a “large” effect size.

ROS: TO-T4
Cohen's d = (0.278 - 0.187) ⁄ 0.082961 = 1.096903.
Glass's delta = (0.278 - 0.187) ⁄ 0.066 = 1.378788.
Hedges' g = (0.278 - 0.187) ⁄ 0.082961 = 1.096903.

ROS: T1A-T1B
Cohen's d = (0.347 - 0.203) ⁄ 0.130545 = 1.103068.
Glass's delta = (0.347 - 0.203) ⁄ 0.072 = 2.
Hedges' g = (0.347 - 0.203) ⁄ 0.130545 = 1.103068.

ROS: T2A-T2B
Cohen's d = (0.316 - 0.191) ⁄ 0.097624 = 1.280419.
Glass's delta = (0.316 - 0.191) ⁄ 0.07 = 1.785714.
Hedges' g = (0.316 - 0.191) ⁄ 0.097624 = 1.280419.

ROS: T3A-T3B
Cohen's d = (0.337 - 0.225) ⁄ 0.090941 = 1.231569.
Glass's delta = (0.337 - 0.225) ⁄ 0.08 = 1.4.
Hedges' g = (0.337 - 0.225) ⁄ 0.090941 = 1.231569.

TAC: T0-T3A
Cohen's d = (0.337 - 0.225) ⁄ 0.090941 = 1.231569.
Glass's delta = (0.337 - 0.225) ⁄ 0.08 = 1.4.
Hedges' g = (0.337 - 0.225) ⁄ 0.090941 = 1.231569.

TAC: T1A-T1B
Cohen's d = (2.4 - 2.894) ⁄ 0.32155 = 1.536308.
Glass's delta = (2.4 - 2.894) ⁄ 0.242 = 2.041322.
Hedges' g = (2.4 - 2.894) ⁄ 0.32155 = 1.536308.

IL-6: TO-T4
Cohen's d = (3.071 - 1.651) ⁄ 1.159105 = 1.225083.
Glass's delta = (3.071 - 1.651) ⁄ 1.005 = 1.412935.
Hedges' g = (3.071 - 1.651) ⁄ 1.159105 = 1.225083.

IL-6: T2A-T2B
Cohen's d = (4.38 - 2.042) ⁄ 1.407579 = 1.661008.
Glass's delta = (4.38 - 2.042) ⁄ 1.034 = 2.261122.
Hedges' g = (4.38 - 2.042) ⁄ 1.407579 = 1.661008.

IL-6: T3A-T3A
Cohen's d = (4.809 - 2.058) ⁄ 1.366362 = 2.013375.
Glass's delta = (4.809 - 2.058) ⁄ 0.926 = 2.970842.
Hedges' g = (4.809 - 2.058) ⁄ 1.366362 = 2.013375.

NEOPTERIN: T1A-T1B
Cohen's d = (217.6 - 116.6) ⁄ 57.229402 = 1.764827.
Glass's delta = (217.6 - 116.6) ⁄ 27.91 = 3.618775.
Hedges' g = (217.6 - 116.6) ⁄ 57.229402 = 1.764827.

NEOPTERIN: T2A-T2B
Cohen's d = (197.8 - 136.5) ⁄ 47.293349 = 1.296165.
Glass's delta = (197.8 - 136.5) ⁄ 44.23 = 1.385937.
Hedges' g = (197.8 - 136.5) ⁄ 47.293349 = 1.296165.

 NEOPTERIN: T3A-T3B
Cohen's d = (240.3 - 164.2) ⁄ 59.946366 = 1.269468.
Glass's delta = (240.3 - 164.2) ⁄ 57.06 = 1.333684.
Hedges' g = (240.3 - 164.2) ⁄ 59.946366 = 1.269468.

URIC ACID: T0-T4
Cohen's d = (41.9 - 9.3) ⁄ 20.134175 = 1.619138.
Glass's delta = (41.9 - 9.3) ⁄ 4.6 = 7.086957.
Hedges' g = (41.9 - 9.3) ⁄ 20.134175 = 1.619138.

UREA: T0-T4
Cohen's d = (736.9 - 948) ⁄ 348.960062 = 0.60494.
Glass's delta = (736.9 - 948) ⁄ 346.3 = 0.609587.
Hedges' g = (736.9 - 948) ⁄ 348.960062 = 0.60494.

K: T2A-T2B
Cohen's d = (92 - 60.5) ⁄ 34.554377 = 0.911607.
Glass's delta = (92 - 60.5) ⁄ 35.1 = 0.897436.
Hedges' g = (92 - 60.5) ⁄ 34.554377 = 0.911607.

P: T3A-T3B
Cohen's d = (54.2 - 105.7) ⁄ 59.016608 = 0.872636.
Glass's delta = (54.2 - 105.7) ⁄ 51.6 = 0.998062.
Hedges' g = (54.2 - 105.7) ⁄ 59.016608 = 0.872636.

TQR: T0-T1
Cohen's d = (240.3 - 164.2) ⁄ 59.946366 = 1.269468.
Glass's delta = (240.3 - 164.2) ⁄ 57.06 = 1.333684.
Hedges' g = (240.3 - 164.2) ⁄ 59.946366 = 1.269468.

TQR: T0-T2
Cohen's d = (240.3 - 164.2) ⁄ 59.946366 = 1.269468.
Glass's delta = (240.3 - 164.2) ⁄ 57.06 = 1.333684.
Hedges' g = (240.3 - 164.2) ⁄ 59.946366 = 1.269468.

TQR: T0-T3
Cohen's d = (240.3 - 164.2) ⁄ 59.946366 = 1.269468.
Glass's delta = (240.3 - 164.2) ⁄ 57.06 = 1.333684.
Hedges' g = (240.3 - 164.2) ⁄ 59.946366 = 1.269468.

SBP (mmHg): T1-T4
Cohen's d = (116.5 - 135.6) ⁄ 19.57575 = 0.975697.
Glass's delta = (116.5 - 135.6) ⁄ 24.9 = 0.767068.
Hedges' g = (116.5 - 135.6) ⁄ 19.57575 = 0.975697.

-       So it should be better to include the scale of Hopkins to interpret the results. 

Thank you for your suggestion. We added in the text

-       Poor resolution of figures. Please change the resolution of at least 300 dpi
We changed the figures resolution, all TIFF at 300dpi

 Discussion.

-       The discussion can be tightened up. As is, it lacks focus in many areas for example the specific purpose and findings of this study 

-       More notes about practical application associated to the results of the study could be implemented. 
The discussion has been in part tightened up an better written.

Conclusion.

The conclusion to be reformulated (please try to make it shorter).
The limitations have been moved in discussion

Reviewer 2 Report

Dear authors

  In the current study, you hypothesized that freeskiing in an extreme environment with significant physical effort can induce reactive oxygen species (ROS) generation and dehydration. Thus youaimed to investigate the evolution of the oxy-inflammation and hydration status during a freeskiing training season through measurement of ROS, total antioxidant capacity (TAC), interleukin-6 (IL-6), 21 nitric oxide (NO) derivatives, neopterin and electrolytes balance changes in blood and saliva collected at T0, 20 before (A) and after (B) T1-T3 and at T4.

However,

-        Introduction is too long and should be reduced.

-        Sample size is very small (8 participants only)

-        Authors did not clarify how they calculate the sample size.

-        Study design lack the presence of a negative control group.

-        You mentioned in the abstract levels in the blood while in methods you measured them in urine!!!

-        Why did you choose to measure the lab. Markers in saliva and urine only??

-        I totally disagree with building conclusions on research on urine samples only. Urine levels of these biomarkers are not accurate as serum levels.

-        Some methods are mentioned without references.

-        Authors should clarify why they choose saliva and urine samples as this is a major limitation of the study (not to measure these markers in blood)??

-        Reference number 41 detects NO in blood. How it is used to detect NO in urine samples.

-        A lot of references are very old and need to be up-to-date.

-        Other minor comments are within the manuscript file.

-         

Author Response

Dear authors

In the current study, you hypothesized that freeskiing in an extreme environment with significant physical effort can induce reactive oxygen species (ROS) generation and dehydration. Thus youaimed to investigate the evolution of the oxy-inflammation and hydration status during a freeskiing training season through measurement of ROS, total antioxidant capacity (TAC), interleukin-6 (IL-6), 21 nitric oxide (NO) derivatives, neopterin and electrolytes balance changes in blood and saliva collected at T0, 20 before (A) and after (B) T1-T3 and at T4.

However,

-        Introduction is too long and should be reduced.

Thanks, the introduction has been reduced and the aim clarified

-        Sample size is very small (8 participants only)

-        Authors did not clarify how they calculate the sample size.

It was not possible to calculate the sample size, because we studied all the athletes of the Prato nevoso freeskers team. This sentence has been added in the text in subjects 2.1 paragraph.

-        Study design lack the presence of a negative control group.

The aims of the study were to: 1) evaluate the effect of a single section of training T1, T2 and T3 at A (pre) and B (post); 2) evaluate the changes during the season. For the first case we previously reported no changes during 3 h at rest (Mrakic Sposta et al. 2012). So we think that a negative control group was not necessary. For the second case we would have to recruit athletes that along a whole season do not practice any physical activity. Very difficult indeed!

-        You mentioned in the abstract levels in the blood while in methods you measured them in urine!!!

Sorry for the blunde. The text has been corrected.

-        Why did you choose to measure the lab. Markers in saliva and urine only??

An explanation has been added in the introduction.

-        I totally disagree with building conclusions on research on urine samples only. Urine levels of these biomarkers are not accurate as serum levels.

 The collection of blood samples in extreme environments isn’t easy and, in some cases, impossible. The assessment of creatinine, uric acid and electrolytes is a routine clinical procedure. Neopterin is usually assessed in urine (Mrakic-Sposta et al 2015, 2019, 2020). IL 6 was previously assessed both in plasma/serum and urine of same subjects (van Oers et al. 1988, Newstead et al 1993, Fan et al 2012, Mrakic-Sposta et al 2015).

-        Some methods are mentioned without references.

The reference has been added.

-        Authors should clarify why they choose saliva and urine samples as this is a major limitation of the study (not to measure these markers in blood)??

An explanation has been added in the introduction.

-        Reference number 41 detects NO in blood. How it is used to detect NO in urine samples.

The reference has been changed.

-        A lot of references are very old and need to be up-to-date.

The reference were reduced and updated.

-        Other minor comments are within the manuscript file.

Thanks.

Reviewer 3 Report

This study determined salivary (reactive oxygen species and total antioxidant capacity) and urinary (interleuin-6, NO metabolites, creatine, uric acid, electrolytes, neopterin, pH, urobilinogen, bilirubin, ketones, density and leukocytes) biomarkers before, during and after a freeskiing training season in amateur freeskiers. The main study limitation is the small sample size (n = 8) that precluded the use of parametric statistical tests and more robust conclusions.

Introduction

The authors did a good job linking the ski modality with the oxidative stress condition and dehydration. However, there remains some aspects that could be clarified with the inclusion of a hypothesis for the expected outcomes. What are the expected outcomes regarding the reactive oxygen species production and the total antioxidant capacity? Why? What are the expected outcomes regarding hydration status? Are the biomarkers levels expected to increase as a cumulative effect of the training sessions?

Finally, why using non-invasive methods? Are these methods valid and reliable? Please, justify it in the introduction.

Methods

The study was well designed, conducted and described. The authors adopted good practices for monitoring the training load by using the subject perceived exertion (BORG), total quality of recovery scale, visual analog scale and profile of mood states.

Was the sample power analyzed?

Results

The presentation of results is adequate.

Was the training adherence registered? Please, include the data if it was registered or justify if it was not.

Discussion

The discussion was well written.

The study has some limitations that it is worth mentioning, besides the small sample. Amateur athletes may not have a training routine that is sufficient to reach the highest performance. Therefore, the results of the present study should not be extrapolated to professional ski athletes. Additionally, the sample may be heterogeneous in terms of physical conditioning due to the amateur level and individual training history. Thus, without an individual control of the exercise intensity is not possible to assume that the volunteers had the exact same stimuli during the training period, even with the subjective perceived exertion been assessed after every training session. If the authors agree, I suggest that the amateur level of the volunteers and the absence of physical assessment and training individualization might also be considered as study limitations.

Conclusion

The conclusion should confirm or refuse the study hypothesis.

Author Response

Comments and Suggestions for Authors

This study determined salivary (reactive oxygen species and total antioxidant capacity) and urinary (interleuin-6, NO metabolites, creatine, uric acid, electrolytes, neopterin, pH, urobilinogen, bilirubin, ketones, density and leukocytes) biomarkers before, during and after a freeskiing training season in amateur freeskiers. The main study limitation is the small sample size (n = 8) that precluded the use of parametric statistical tests and more robust conclusions.

Introduction

The authors did a good job linking the ski modality with the oxidative stress condition and dehydration. However, there remains some aspects that could be clarified with the inclusion of a hypothesis for the expected outcomes.

What are the expected outcomes regarding the reactive oxygen species production and the total antioxidant capacity? Why?

No hypothesis are possible as the freeski activity has never been analyzed before. A sentence was added in the introduction. Moreover, in introduction is reported: Physical exercise is a well-known source of reactive oxygen species (ROS) in response to raised O2 utilization to sustain the effort

What are the expected outcomes regarding hydration status?

Are the biomarkers levels expected to increase as a cumulative effect of the training sessions?

In introduction is reported:

It is well known that physical training improves the efficiency of endogenous antioxidant system reducing exercise-induced oxidative stress. However, prolonged training session can reduce the efficiency of antioxidant defenses leading to an excessive ROS production.

Finally, why using non-invasive methods? Are these methods valid and reliable? Please, justify it in the introduction.

 A sentence has been added in the introduction.

Methods

The study was well designed, conducted and described. The authors adopted good practices for monitoring the training load by using the subject perceived exertion (BORG), total quality of recovery scale, visual analog scale and profile of mood states.

Was the sample power analyzed?

It was not possible to calculate the sample size, because we studied all the athletes of the Prato nevoso freeskers team. This sentence has been added in the text in subjects 2.1 paragraph.

Results

The presentation of results is adequate.

Was the training adherence registered? Please, include the data if it was registered or justify if it was not.

 In every training session, all the volunteers respected the training plan and the timing. During the warm-up, freeskiers subjects ran across the ski track (500 meters length) close to snowpark three times (5 minutes per time) as follows:

-          1st 100 m: 2 Nose Butter 360 and 1 Nose Butter 540

-          2nd 100 m: 3 Switch tail butter 360

-          3th 100 m: 2 Nose Butter 360 and 2 Nose Butter 540.

-          4th 100 m: 2 Nollie and 2 Nollie 180

-          5th 100 m: 2 Switch tail butter 360, 1 Hand drag 180 and 1 Hand Drag 360.

The chairlift takes 9 minutes to reach the top of the snow park. Always, Freeskiers used the ski track number 2.

After warm-up, all the volunteers ride the snow park line of Prato Nevoso Ski resort six times (5 minutes per time). The length of snow park line was 500 m. 

As described above, snow park is close to ski track: for this reason, volunteers used the same chairlift.

We have added in the paragraph: 2.2.1. Training exercise session some sentences.

Discussion

The discussion was well written.

The study has some limitations that it is worth mentioning, besides the small sample. Amateur athletes may not have a training routine that is sufficient to reach the highest performance. Therefore, the results of the present study should not be extrapolated to professional ski athletes. Additionally, the sample may be heterogeneous in terms of physical conditioning due to the amateur level and individual training history. Thus, without an individual control of the exercise intensity is not possible to assume that the volunteers had the exact same stimuli during the training period, even with the subjective perceived exertion been assessed after every training session. If the authors agree, I suggest that the amateur level of the volunteers and the absence of physical assessment and training individualization might also be considered as study limitations.

This sentence has been added as limitation.

Conclusion

The conclusion should confirm or refuse the study hypothesis.

In the conclusions is reported:

Freeskiing induced an increase of ROS production that can be contained by antioxidant defense activation. Neopterin raised as consequence of ROS generation while IL-6 as consequence of physical activity and muscular contraction type. We did not find deep changes in electrolytes balance probably because all subjects examined were well trained and very experienced.
